Accepted to the NeurIPS 2024 Workshop on Safe Generative AI

# Variational Diffusion Unlearning: A Variational Inference Framework for Unlearning in Diffusion Models

**Subhodip Panda**[1], **M S Varun**[*2], **Shreyans Jain**[*3], **Sarthak Kumar Maharana**[4] **& Prathosh A.P.**[1]

[1] Department of ECE, Indian Institute of Science, Bangalore, India
[2] Department of Computer Science, PES University, Bangalore, India
[3] Department of EE, Indian Institute of Technology, Gandhinagar, India
[4] Department of Computer Science, The University of Texas at Dallas, TX, USA
{subhodipp, prathosh}@iisc.ac.in, varun80042@gmail.com,
  shreyans.jain@iitgn.ac.in, sarthak.maharana@utdallas.edu

## Abstract

For responsible and safe deployment of diffusion models in various domains, regulating the generated outputs from these models is desirable because such models could generate undesired violent and obscene outputs. To tackle this problem, one of the most popular methods is to use *machine unlearning* methodology to forget training data points containing these undesired features from pre-trained generative models. Thus, the principal objective of this work is to propose a machine unlearning methodology that can prevent the generation of outputs containing undesired features from a pre-trained diffusion model. Our method termed as Variational Diffusion Unlearning (**VDU**) is a **one-step method** that *only requires access to a subset of training data containing undesired features to forget*. Our approach is inspired by the variational inference method that minimizes a loss function consisting of two terms: *plasticity inducer* and *stability regularizer*. *Plasticity inducer* reduces the log-likelihood of the undesired training data points while the *stability regularizer*, essential for preventing loss of image sample quality, regularizes the model in parameter space. We validate the effectiveness of our method through comprehensive experiments, by forgetting data of certain user-defined classes from MNIST and CIFAR-10 datasets from a pre-trained unconditional denoising diffusion probabilistic model (DDPM).

## 1 Introduction

*(a) Original "1"*   *(b) Unlearned "1"*   *(c) Original "frog"*   *(d) Unlearned "frog"*

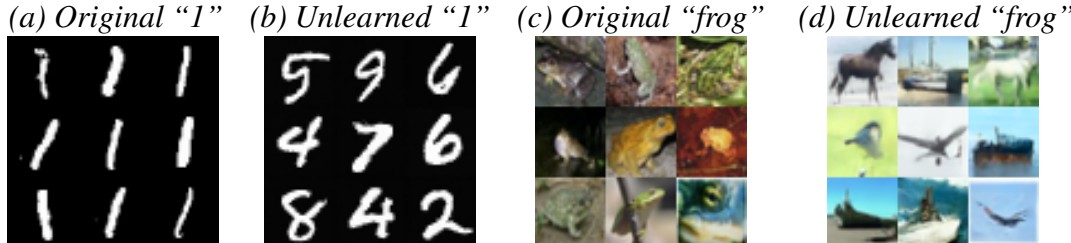

Figure 1: (a) and (c) show the original images generated by a pre-trained DDPM model on the MNIST and CIFAR-10 datasets, respectively. (b) and (d) display the corresponding images generated after unlearning, using our method **VDU**. The same noise vectors used to generate the original images were applied in the unlearned model to generate the unlearned images. **VDU** delivers good-quality images after unlearning, as well.

---

*denotes equal contribution

In recent years, diffusion models (Ho et al., 2020; Song & Ermon, 2019; Song et al., 2021; Rombach et al., 2022) have been popular for generating high-quality images which are useful for various tasks such as image and video editing (Ceylan et al., 2023; Feng et al., 2024), text-to-image translation (Ramesh et al., 2021; 2022; Saharia et al., 2022) etc. As these models become more and more widespread, there lies a requirement to train them on vast amounts of internet data for diverse and robust output generation. However, there is also a potential downside to using such models, as they often generate outputs containing biased, violent, and obscene features (Tommasi et al., 2017). Thus, a safe and responsible generation from these models becomes an important requirement.

To address these challenges, recent works (Moon et al., 2024; Tiwary et al., 2023; Panda & A.P., 2023; Gandikota et al., 2023; Schramowski et al., 2023; Heng & Soh, 2023) have proposed methods for regulating the outputs of various generative models (e.g. VAEs (Kingma, 2013), GANs (Goodfellow et al., 2020), and Diffusion models (Ho et al., 2020)) to ensure their safe and responsible deployment, with *machine unlearning* emerging as a particularly important technique for controlling the safe generation of content from these models. The key idea of *machine unlearning* is to develop a computationally efficient method to forget the subset of training data containing these undesired features from this pre-trained model. Thus, ideally, the *unlearned* model should behave like the retrained model — trained without the undesired data subset. However, achieving this goal is challenging because, during the process of unlearning, the model's generalization capacity gets hurt making the quality of the generated outputs poor. This phenomenon is well studied in a similar context also known as *catastrophic forgetting* (McCloskey & Cohen., 1989; Goodfellow et al., 2014; Kirkpatrick et al., 2017; Ginart et al., 2019; Nguyen et al., 2020a) where plasticity to adapt to a new task hurts the stability of the model to perform well on the older task it had been trained on. In this case, the new task of unlearning an undesired subset of data hurts the quality of images generated by the unlearned model thereafter. Thus, the scope of this research is concerned with the following question:

*Can we develop a machine unlearning algorithm that can forget an undesired subset of training data from a pre-trained diffusion model without hurting the quality of the images it generates?*

To answer this question, a recent unlearning work Selective Amnesia (SA) (Heng & Soh, 2023) adopts a continual learning setup and proposes an unlearning method based on elastic weight consolidation (EWC) (Kirkpatrick et al., 2017) wherein a weight regularization strategy in the parameter space is introduced to balance the unlearning task and retaining sample quality. In essence, the variation of parameters is penalized by computing the "importance", to retain good performance, using the Fisher Information Matrix (FIM). However, this unlearning method has two potential downsides: (a) EWC formulation requires the calculation of the FIM which is expensive due to the gradient product. (b) This unlearning method struggles to maintain the model's performance, often leading to the generation of low-quality samples when it relies solely on the unlearning data subset. To solve the problem of low image quality, the authors employ generative replay to retrain the model with generated samples from "non-unlearning" data subsets, which further enhances computational requirements. A major challenge, however, lies in situations with partial access to the training data due to rising concerns for data privacy and safety (Bae et al., 2018). In such a case, Selective Amnesia (SA) performs poorly with no access to the non-unlearning data.

Taking note of such crucial observations, our research aims to develop a computationally efficient algorithm for unlearning an undesired class of training data from a pre-trained unconditional Denoising Diffusion Probabilistic Model (DDPM) (Ho et al., 2020). It is also important to mention that our methodology only requires partial access to a subset of training data aimed at unlearning because it is not always feasible to have access to the full training dataset (Chundawat et al., 2023b; Panda et al., 2024). While such a realistic setup is challenging, we draw inspiration from works on variational inference techniques (Knoblauch et al., 2022; Nguyen et al., 2018; Noel Loo, 2021; Wild et al., 2022). We develop, Variational Diffusion Unlearning (**VDU**), a variational inference framework in the parameter space to unlearn a subset of training data. This theoretical formulation of the variation divergence yields a lower bound which is used as a loss function to fine-tune the pre-trained model for unlearning. This loss consists of two terms: *plasticity inducer* and *stability regularizer*. The *plasticity inducer* is used for adapting to the new task of reducing the log-likelihood of the unlearning data while the *stability regularizer* prevents drastic changes in the pre-trained parameters of the model. These two proposed terms capture the persistent trade-offs that exist between

the quantity of unlearning required and maintaining initial image quality. Overall, our contributions are summarized as follows:

- We propose a theoretical formulation of unlearning from a variational inference perspective and propose a methodology to unlearn a certain class of training data from a pre-trained unconditional denoising diffusion probabilistic model (DDPM) (Ho et al., 2020).

- To address the limitations of concurrent unlearning methods for diffusion models (Heng & Soh, 2023), which stem from the computational complexity of FIM computation and the need for generative replay with non-unlearned data, our proposed method is more efficient. It achieves computational efficiency by fine-tuning the pre-trained model for only a few epochs—sometimes as few as one—and is effective with fewer samples, requiring access only to the unlearning data points, making it highly suitable for stricter unlearning scenarios with limited access to the original training dataset.

- We validate our method on the MNIST (LeCun et al., 1998) and CIFAR-10 (Krizhevsky et al., 2009) datasets, for unlearning different classes of data points using a pre-trained DDPM.

## 2 RELATED WORKS

### 2.1 MACHINE UNLEARNING FOR GENERATIVE MODELS

The core of machine unlearning (Cao & Yang, 2015; Xu et al., 2020; Nguyen et al., 2022; Bourtoule et al., 2021) revolves around removing or forgetting a specific subset of training data from a trained model, either due to rising privacy and security concerns (Bae et al., 2018), potential fairness (Mehrabi et al., 2021), or as per a user's request. A plausible approach is to retrain the model on the training data devoid of the undesired training data subset. However, this can be computationally very expensive concerning the scale of the model parameters and training data. To solve this problem, different machine unlearning algorithms were proposed for different problem and model settings such as for K-means (Ginart et al., 2019), random forests (Brophy & Lowd, 2021), linear classification models (Guo et al., 2019; Golatkar et al., 2020a;b; Sekhari et al., 2021), neural network based classifiers (Wu et al., 2020; Graves et al., 2021; Chundawat et al., 2023a; Panda et al., 2024) etc.

However, machine unlearning is not only useful for the above supervised and unsupervised learning scenarios but also necessary for generative settings. With the emergence of large pre-trained text-to-image models (Rombach et al., 2022; Saharia et al., 2022), there is potential for misuse in generating harmful or inappropriate content. Thus, controlling the outputs from these generative models becomes an utmost priority. To solve this problem, recent works (Sun et al., 2023; Tiwary et al., 2023; Moon et al., 2024) proposed unlearning-based approaches for variational auto-encoders (VAEs) and generative adversarial networks (GANs). Sun et al. (2023) proposed a cascaded unlearning method using the idea of latent space substitution for a pre-trained StyleGAN under both settings of full and partial access (similar to our setting) to the training dataset. Tiwary et al. (2023) proposed a two-stage *adapt and unlearn* approach of first adapting a pre-trained StyleGAN to undesired samples and then unlearning the model using the regularization in parameter space. Further to extend unlearning for diffusion models, a recent work (Heng & Soh, 2023) adopts a continual learning setup and proposes an unlearning method based on elastic weight consolidation (EWC) (Kirkpatrick et al., 2017) for unlearning a pre-trained conditional DDPM (Ho et al., 2020).

### 2.2 VARIATIONAL INFERENCE

To acquire exact inference from data, it is essential to calculate the exact posterior distribution. However, the exact posterior is often intractable and hard to calculate essentially making the inference task challenging. To solve this problem, the domain of variational inference tries to approximate the true posterior by a more tractable distribution from a class of distributions. Now to get the optimal distribution, often termed as variational posterior, these methods (Sato, 2001; Broderick et al., 2013; Blundell et al., 2015; Bui et al., 2016; Ghahramani & Attias, 2020) optimize the so-called evidence lower bound (ELBO). These methodologies formulate the problem of inference in the parameter or weight space which is often challenging because of the high dimension of the parameter space

and multi-modality of parameter posterior distribution. Thus to solve this problem, the recent line of works (Ma et al., 2019; Sun et al., 2019; Rudner et al., 2020; Wild et al., 2022) try to do inference in the function space itself. These methods (Rudner et al., 2020; Wild et al., 2022) perform inference by optimizing functional KL-divergence, minimizing the Wasserstein distance between the functional prior and Gaussian process. Inspired by these works, we formulate our unlearning methodology from a task of inference in parameter space by minimizing a variational divergence.

## 3 METHODOLOGY

### 3.1 PROBLEM FORMULATION

Consider a pre-trained DDPM model, denoted as $f_{\theta^*}$, with initial parameters $\theta^* \in \Theta \subseteq \mathbb{R}^d$. $\Theta$ denotes the complete parameter space. This model has been trained on a specific training dataset $D$, consisting of $m$ i.i.d. samples $\{x_i\}_{i=1}^m$ that are drawn from a distribution $P_{\mathcal{X}}$ over the data space $\mathcal{X}$, tries to learn the underlying data distribution $P_{\mathcal{X}}$. Based on the outputs of this model, the user wants to unlearn a portion of the data space consisting of undesired features, referred to as $\mathcal{X}_f$. Therefore, the entire data space can be expressed as the union of $\mathcal{X}_r$ and $\mathcal{X}_f$, where $\mathcal{X} = \mathcal{X}_r \bigcup \mathcal{X}_f$ or $\mathcal{X}_r = \mathcal{X} \setminus \mathcal{X}_f$. We denote the distributions over $\mathcal{X}_f$ and $\mathcal{X}_r$ as $P_{\mathcal{X}_f}$ and $P_{\mathcal{X}_r}$, respectively. The objective of the unlearning mechanism is to output a sanitized model $\theta^u$ that does not produce outputs within the domain $\mathcal{X}_f$. This implies that the model should be trained to generate data samples conforming to the distribution $P_{\mathcal{X}_r}$ only. Assuming access to the whole training dataset, a computationally expensive approach to achieving this is by retraining the entire model from scratch using a dataset $D_r = \{x_i\}_{i=1}^s \sim P_{\mathcal{X}_r}^s$ or equivalently, $D_r = D \setminus D_f$, where $D_f = \{x_i\}_{i=1}^t \sim P_{\mathcal{X}_f}^t$. It is important to note that we do not have access to $D_r$ in our setting. Hence, the method of retraining becomes infeasible.

### 3.2 METHOD OVERVIEW

Given the pre-trained DDPM model $f_{\theta^*}$ and unlearning data subset $D_f$, the objective is to produce an unlearned model $\theta^u$ so that it behaves like a retrained model $\theta^r$ trained on $D_r$. Inspired by some previous works in Bayesian inference (Sato, 2001; Broderick et al., 2013; Blundell et al., 2015; Ghahramani & Attias, 2020; Nguyen et al., 2020b), it can be seen that, retrained parameters $\theta^r$ are a sample from the retrained model's parameter posterior distribution $P(\theta|D_r)$ i.e., $\theta^r \sim P(\theta|D_r)$. Similarly, the pre-trained model's parameters $\theta^* \sim P(\theta|D_r, D_f)$. Using this motivation, we try to approximate the retrained model's parameter posterior distribution $P(\theta|D_r)$ as follows:

$$P(\theta|D_r, D_f) \propto P(D_r, D_f|\theta)P(\theta) \propto P(D_f|\theta)P(D_r|\theta)P(\theta) \propto P(D_f|\theta)P(\theta|D_r) \quad (1)$$

Eq. 1 is the direct consequence of Bayes' rule ignoring the normalizing constant and assuming the conditional independence between $D_r$ and $D_f$ given $\theta$. It can be seen from Eq. 1 that the posterior distribution $P(\theta|D_r)$ is intractable and an approximation is required by forming $proj(P(\theta|D_r)) \approx Q^*(\theta)$. Here, *proj*(·) is a projection function that takes an intractable unnormalized distribution and maps it to a normalized distribution. As previously mentioned in the literature (Broderick et al., 2013; Blundell et al., 2015; Bui et al., 2016; Nguyen et al., 2018; Noel Loo, 2021; Knoblauch et al., 2022; Wild et al., 2022), one can take several choices of projection functions such as Laplace's approximation, variational KL-divergence minimization, moment matching, and importance sampling. We adopt variational KL-divergence minimization for our method, as prior research (Bui et al., 2016) has demonstrated its superior performance over other inference techniques for complex models. Thus, our method is defined through a variational KL-divergence minimization over set of probable approximate posterior distribution $\mathcal{Q}$ as follows:

$$Q^*(\theta) = \underset{Q(\theta) \in \mathcal{Q}}{\operatorname{argmin}} \, D_{KL}\left(Q(\theta) \middle\| Z \cdot \frac{P(\theta|D_f, D_r)}{P(D_f|\theta)}\right) \quad (2)$$

Here, $Z$ is the intractable normalization constant which is independent of the parameter $\theta$. Finally, if the variational posterior distribution $Q(\theta) = \prod_{i=1}^d \mathcal{N}(\theta_i, \sigma_i^2)$ and the posterior distribution with

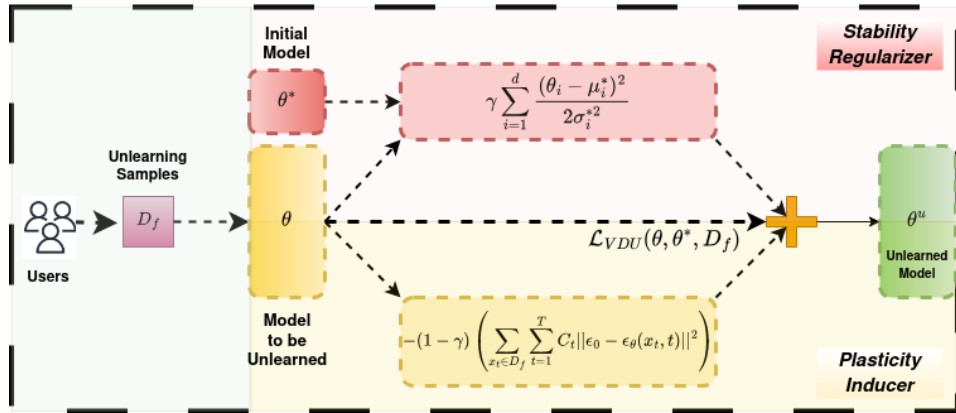

Figure 2: **Variational Diffusion Unlearning (VDU):** Given user-identified samples to be unlearned ($D_f$), our unlearning method fine-tunes the initial pre-trained DDPM model with a loss function having two terms: the first component is a *Plasticity inducer* (bottom half) aiming to minimize the log-likelihood associated with the unlearned data points while the second one is a *Stability regularizer* (upper half) aiming to retain the performance of the model.

full data $P(\theta|D_r, D_f) = \prod_{i=1}^{d} \mathcal{N}(\mu_i^*, \sigma_i^{*2})$, Eq. 2 results in the minimization of the loss function below which we define as the variational diffusion unlearning (VDU) loss as follows:

$$\mathcal{L}_{VDU}(\theta, \theta^*, D_f) = \underbrace{-(1-\gamma)\left(\sum_{x_t \in D_f} \sum_{t=1}^{T} \frac{(1-\alpha_t)}{\alpha_t(1-\bar{\alpha}_{t-1})}||\epsilon_0 - \epsilon_\theta(x_t, t)||^2\right)}_{A} + \underbrace{\gamma \sum_{i=1}^{d} \frac{(\theta_i - \mu_i^*)^2}{2\sigma_i^{*2}}}_{B}$$

(3)

---

**Algorithm 1** Variational Diffusion Unlearning (VDU)

**Input**: unlearning data: $D_f$, initial parameter: $\theta^*$, no. of epochs: $E$, learning rate: $\eta$ and, hyper-parameter: $\gamma$

    Initialize: $\theta \leftarrow \theta^*$
    **for** $t = 1$ to $E$ **do**
        $\mathcal{L}_{VDU}(\theta, \theta^*, D_f) = -(1-\gamma)\left(\sum_{x_t \in D_f} \sum_{t=1}^{T} \frac{(1-\alpha_t)}{\alpha_t(1-\bar{\alpha}_{t-1})}||\epsilon_0 - \epsilon_\theta(x_t, t)||^2\right) + \gamma \sum_{i=1}^{d} \frac{(\theta_i - \mu_i^*)^2}{2\sigma_i^{*2}}$
        $\theta^{t+1} \leftarrow \theta^t - \eta \nabla_\theta \mathcal{L}_{VDU}(\theta, \theta^*, D_f)$
    **end for**
    **Output**: $\theta^E$

---

Eq. 3 represents the proposed loss function used to optimize the pre-trained model during the unlearning process. The loss comprises two key terms: $A$ term referred to as the "plasticity inducer," minimizes the log-likelihood of the unlearning data, while $B$ term serves as the "stability regularizer," penalizing the model's parameters to prevent them from deviating too much from their pre-trained state during unlearning. To balance these two components, we introduce a hyper-parameter, $\gamma$. $\{\alpha_t : t \in T\}$ refers to the diffusion model's noise scheduler where $\bar{\alpha}_t = \prod_{j=1}^{t} \alpha_j$. $\epsilon_0$ is the true added noise, $\epsilon_\theta(x_t, t)$ is the model predicted noise at time $t$ and, $d$ is the dimension of parameter. Figure 2 is a detailed illustration of our framework and details the different loss components used for the unlearning of $D_f$.

### 3.3 THEORETICAL OUTLOOK

In this section, we provide theoretical exposition for the derivation of the variational diffusion unlearning loss $\mathcal{L}_{VDU}(\theta, \theta^*, D_f)$ in Eq. 3 from variational divergence minimization defined in

Eq. 2. Let $\{x_t : t = 0, 1, \ldots, T\}$ denote latent variables with $x_0$ denoting the true data. It is important to mention that in the diffusion process, it is assumed that all transitional kernels are first-order Markov. Now, in the forward diffusion process, the transition kernel is denoted as $q(x_t|x_{t-1})$ with the joint posterior distribution being $q(x_{1:T}|x_0) = \prod_{t=1}^{T} q(x_t|x_{t-1})$ where each $q(x_t|x_{t-1}) = \mathcal{N}(x_t; \sqrt{\alpha_t}x_{t-1}, (1-\alpha_t)I)$. Similarly, for the backward diffusion process, the transitional kernel is denoted as $p(x_{t-1}|x_t)$ with joint distribution $p(x_{0:T}) = p(x_T) \prod_{t=1}^{T} p_\theta(x_{t-1}|x_t)$ where, $p(x_T) = \mathcal{N}(x_T; 0, I)$. Thus after optimizing the diffusion model, the sampling procedure is done by sampling Gaussian noise from $p(x_T)$ and iteratively running the denoising transitions $p_\theta(x_{t-1}|x_t)$ for $T$ steps to generate a new sample $x_0$.

**Lemma 1** *Assuming all the transition kernels to be Gaussian, the following holds:*

*I.* $q(x_{t-1}|x_t, x_0) = \mathcal{N}(x_{t-1}; \mu_q(t), \sigma_q^2(t)I)$ *with* $\mu_q(t) = \frac{1}{\sqrt{\alpha_t}}x_t - \frac{1-\alpha_t}{\sqrt{1-\bar{\alpha}_t}\sqrt{\alpha_t}}\epsilon_0$

*II.* $p_\theta(x_{t-1}|x_t) = \mathcal{N}(x_{t-1}; \mu_\theta(t), \sigma_q^2(t)I)$ *with* $\mu_\theta(t) = \frac{1}{\sqrt{\alpha_t}}x_t - \frac{1-\alpha_t}{\sqrt{1-\bar{\alpha}_t}\sqrt{\alpha_t}}\epsilon_\theta(x_t, t)$

*III.* $\sigma_q^2(t) = \frac{(1-\alpha_t)(1-\bar{\alpha}_{t-1})}{(1-\bar{\alpha}_t)}$

**Lemma 2** *The log-likelihood under the backward diffusion process kernel,*

$$\ln p_\theta(x_0) \gtrsim -\sum_{t=2}^{T} \mathop{\mathbb{E}}_{q(x_t|x_0)} [D_{KL}(q(x_{t-1}|x_t, x_0)||p_\theta(x_{t-1}|x_t))]$$

Partial derivation for the above two lemmas can be found in Luo (2022). For completeness, we have added the full proof in Appendix 6.1.1 and 6.1.2.

**Theorem 1** *Assuming a Gaussian mean-field approximation in the parameter space i.e., if the variational posterior distribution $Q(\theta) = \prod_{i=1}^{d} \mathcal{N}(\theta_i, \sigma_i^2)$ and the posterior distribution with full data $P(\theta|D_r, D_f) = \prod_{i=1}^{d} \mathcal{N}(\mu_i^*, \sigma_i^{*2})$ then,*

$$D_{KL}\left(Q(\theta)\middle\|Z \cdot \frac{P(\theta|D_f, D_r)}{P(D_f|\theta)}\right) \gtrsim \underbrace{-\sum_{x_t \in D_f}\sum_{t=2}^{T} \frac{(1-\alpha_t)}{\alpha_t(1-\bar{\alpha}_{t-1})}||\epsilon_0 - \epsilon_\theta(x_t,t)||^2}_{I}$$

$$+ \underbrace{\sum_{i=1}^{d}\left[\frac{(\theta_i - \mu_i^*)^2}{2\sigma_i^{*2}} + \frac{\sigma_i^2}{2\sigma_i^{*2}} + \log\frac{\sigma_i^*}{\sigma_i} - \frac{1}{2}\right]}_{II}$$

**Proof 1** *Here we give a sketch of the proof. For a detailed proof, please look into Appendix 6.1.3. The KL-divergence term in Eq. 2 is expanded and segregated into two terms: $\mathbb{E}[\log P(D_f|\theta)]$ and $D_{KL}(Q(\theta)||P(\theta|D_r, D_f))$. The first term is approximated using Lemmas 1 and 2, while the second term is expanded using the assumption of KL divergence between two Gaussian distributions.*

**Remark 1** *In Theorem 1, the first term $I$ appears in the first part of loss function $\mathcal{L}_{VDU}(\theta, \theta^*, D_f)$ in Eq. 3. While if we assume $\sigma_i = \sigma_i^*$ term $II$ turns out to be simply $\sum_{i=1}^{d}[\frac{(\theta_i-\mu_i^*)^2}{2\sigma_i^{*2}}]$, which is the second component of our proposed loss function.*

## 4 EXPERIMENTS AND RESULTS

### 4.1 DATASETS AND MODELS

The primary goal of our unlearning method is to stop the generation of undesired images from a pre-trained DDPM model. We utilize two well-known datasets for our experiments: MNIST (LeCun et al., 1998) and CIFAR-10 (Krizhevsky et al., 2009). Here, we use unconditional DDPM models for

our unlearning method. We use two different U-Net architectures for the DDPM model on MNIST and CIFAR-10 respectively. These architectures are used from two open-source implementations detailed in Appendix 6.2.1.

## 4.2 Initial Training, Unlearning and Baseline

• **Initial Training:** We train the unconditional DDPM model on MNIST for 40 epochs with a batch size of 64 to obtain the pre-trained model, which achieves an FID (Heusel et al., 2017) score of 5.12. Similarly, to obtain the pre-trained model for the CIFAR-10 dataset, we used a pre-trained check-point from an open-source implementation and fine-tuned the model for a further 90,000 iterations with a batch size of 128 to achieve an FID score of 7.96. A more detailed description of the initial training setups can be found in Appendix 6.2.2.

• **Unlearning:** For MNIST, we unlearn the digit classes **0**, **1**, and **8**. Similarly, for the CIFAR-10 dataset, we target the unlearning of specific classes: **class 1 (automobile)**, **class 6 (frog)**, and **class 8 (ship)**. As can be seen from our method, we require the model parameter's mean and variance, so we have trained 5 models on each dataset to calculate $\mu_i^*$ and $\sigma_i^*$. Further experimental details of our unlearning method on each dataset are added in Appendix 6.2.3.

• **Baseline:** For comparison, we have adapted the state-of-the-art Selective Amnesia (SA) (Heng & Soh, 2023) as the baseline, as it is the most relevant to our approach. A detailed comparison can be found in Table 1. This baseline method relies on a computationally expensive generative replay technique, essentially retraining the model to preserve the quality of generated samples from the unlearned model. In contrast, our approach eliminates the need for such retraining, offering a more efficient alternative.

## 4.3 Evaluation Metrics

To evaluate different unlearning methods for generative models, previous work (Tiwary et al., 2023) proposed below metrics which are described as follows:

• **Percentage of Unlearning (PUL):** This metric measures how much unlearning has occurred by comparing the reduction in the number of unwanted samples produced by the DDPM model after unlearning ($\theta^u$) with the number of such samples before unlearning ($\theta^*$). The Percentage of Unlearning (PUL) is calculated as: $\text{PUL} = \frac{(D_f^g)_{\theta^*} - (D_f^g)_{\theta^u}}{(D_f^g)_{\theta^*}} \times 100\%$ where $(D_f^g)_{\theta^*}$ and $(D_f^g)_{\theta^u}$ represent the number of undesired samples generated by the original DDPM and the unlearned DDPM, respectively. To calculate PUL, we generate 5,000 random samples from both DDPM models and use a pre-trained classifier to identify the unwanted samples.

• **Unlearned Fréchet Inception Distance (u-FID):** To evaluate that the unlearning model doesn't render the pre-trained model useless i.e., to quantify the quality of generated images by the unlearned DDPM, we utilize the u-FID score. It is important to mention that this FID score is measured between the generated samples from the unlearned model and the real data only consisting of non-unlearning data. Thus, to remove the unlearning data points from the real data we use a pre-trained classifier. In this case, a lower u-FID score reflecting higher image quality indicates that the unlearned model's performance does not degrade on the non-unlearning data points.

## 4.4 Experimental Results

Table 1 shows the performance of our method compared to the Selective Amnesia (SA) baseline for different class unlearning settings. It is observable that our method achieves lower u-FID scores with superior to comparable PULs, offering a more favorable trade-off than the SA method. On the MNIST dataset, our method outperforms the SA method (which was trained for 2 epochs) after just 1 epoch of training. However, for CIFAR-10, the SA method achieved its best results with 2 epochs with poor sample quality (see Appendix 6.3.2), while our method required 4 epochs to reach optimal performance and maintain good sample quality. Table 2 further shows the performance of our method for different values of $\gamma$.

In Figure 3, we illustrate the visual performance of our method for different class unlearning scenarios on both MNIST and CIFAR-10 datasets. Further visual results are illustrated in Appendix 6.3.

Table 1: Quantitative performance comparison on MNIST and CIFAR-10 datasets.

| Datasets | Unlearned Classes | Selective Amnesia PUL(%) | u-FID | VDU (Our Method) PUL(%) | u-FID |
|---|---|---|---|---|---|
| MNIST | Digit-0 | **77.47** | 121.61 | 61.00 | **29.33** |
| | Digit-1 | 15.01 | 301.25 | **75.06** | **14.42** |
| | Digit-8 | 48.95 | 161.08 | **68.96** | **38.20** |
| CIFAR-10 | Automobile | 2.25 | 92.89 | **60.87** | **30.85** |
| | Frog | **66.39** | 111.95 | 62.56 | **30.17** |
| | Ship | **85.02** | 249.88 | 71.63 | **24.45** |

Table 2: Unlearning performance with different values of $\gamma$.

| Datasets and Classes | $\gamma$ | VDU (Our Method) PUL(%) | u-FID |
|---|---|---|---|
| MNIST Digit-1 | 0.1 | 75.06 | 14.43 |
| | 0.3 | 72.15 | 14.33 |
| | 0.6 | 55.21 | 11.54 |
| | 0.8 | 71.67 | 13.12 |
| CIFAR-10 Ship | 0.1 | 71.63 | 24.46 |
| | 0.3 | 74.65 | 28.75 |
| | 0.6 | 56.54 | 20.51 |
| | 0.8 | 69.95 | 19.64 |

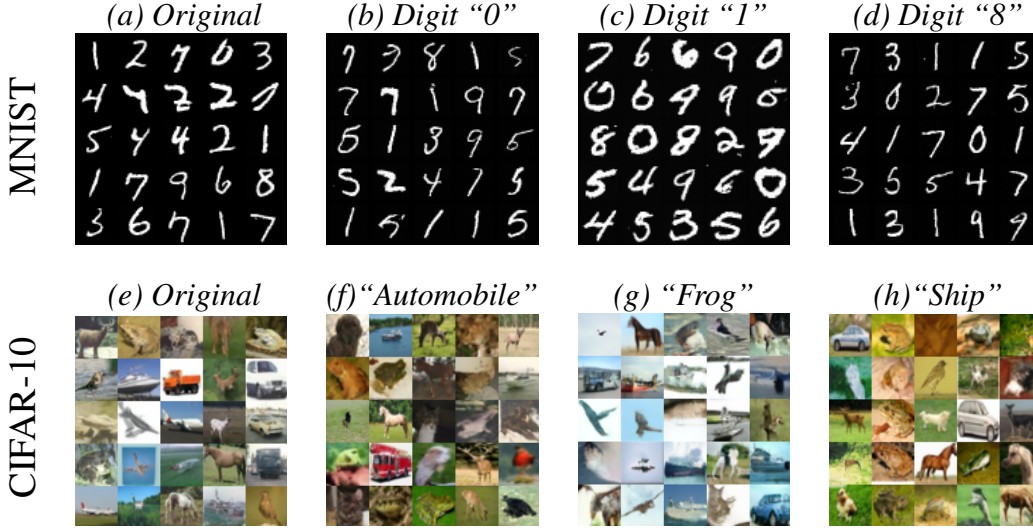

Figure 3: Generated samples from the pre-trained DDPM model and different class unlearned models. (a),(b),(c), and (d) in the first row are generated using pre-trained and unlearned models respectively trained on MNIST while (e),(f),(g), and (h) in the second row images are generated using initial and unlearned models trained on CIFAR-10. Our method shows superior image quality after unlearning.

## 5  CONCLUSION, LIMITATIONS, AND FUTURE WORKS

For the safe and responsible deployment of generative models, it is essential to regulate outputs that contain undesired features. This work presents a machine unlearning methodology to prevent the generation of undesired outputs from a pre-trained unconditional denoising diffusion probabilistic model (DDPM) without accessing the whole training data. Our method termed as Variational Diffusion Unlearning (**VDU**) presents a variational inference framework in parameter space to reduce undesired number of sample generation effectively with a lower computational cost. We show the effectiveness of our method on different class unlearning settings for lower dimensional datasets such as MNIST and CIFAR-10. Acknowledging limited experimental evidence of **VDU** only on lower dimensional datasets, our current and future efforts are as follows:

• **Future experiments:** To show further experimental evidence for the effectiveness of our method, we plan to test our method for high-dimensional datasets such as miniImageNet (Vinyals et al., 2016) and CelebA (Liu et al., 2015).

• **Theoretical Generalization:** Our current theoretical framework leverages the parameter space to exploit variational inference, but its scope is limited. Inspired by the idea of function space variational inference techniques (Ma et al., 2019; Rudner et al., 2020; Sun et al., 2019; Wild et al., 2022), our current efforts also involve finding a superior variational inference framework for machine unlearning.

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

## 6 APPENDIX

### 6.1 THEORETICAL PROOFS

#### 6.1.1 PROOF OF LEMMA-1

Even though it is a three-part proof part-*III* comes from proof of part-*I* while part-*II* is proved via a similar argument as proof of part-*I*. So here we will prove part-*I* in detail. We know:

$$q(x_t|x_{t-1}, x_0) = q(x_t|x_{t-1}) = (x_t; \sqrt{\alpha_t} x_{t-1}, (1-\alpha_t)I) \tag{4}$$

Now, we can represent $x_t$ in terms of $x_0$ by recursive re-parameterization as,

$$x_t = \sqrt{\alpha_t} x_{t-1} + \sqrt{1-\alpha_t} \epsilon_{t-1} \tag{5}$$

$$= \sqrt{\alpha_t} \left( \sqrt{\alpha_{t-1}} x_{t-2} + \sqrt{1-\alpha_{t-1}} \epsilon_{t-2} \right) + \sqrt{1-\alpha_t} \epsilon_{t-1} \tag{6}$$

$$= \sqrt{\alpha_t \alpha_{t-1}} x_{t-2} + \sqrt{\alpha_t - \alpha_t \alpha_{t-1}} \epsilon_{t-2} + \sqrt{1-\alpha_t} \epsilon_{t-1} \tag{7}$$

$$= \sqrt{\alpha_t \alpha_{t-1}} x_{t-2} + \sqrt{\left(\sqrt{\alpha_t - \alpha_t \alpha_{t-1}}\right)^2 + \sqrt{(1-\alpha_t)^2}} \epsilon_{t-2} \tag{8}$$

$$= \sqrt{\alpha_t \alpha_{t-1}} x_{t-2} + \sqrt{\alpha_t - \alpha_t \alpha_{t-1} + 1 - \alpha_t} \epsilon_{t-2} \tag{9}$$

$$= \sqrt{\alpha_t \alpha_{t-1}} x_{t-2} + \sqrt{1 - \alpha_t \alpha_{t-1}} \epsilon_{t-2} \tag{10}$$

$$\vdots \tag{11}$$

$$= \sqrt{\prod_{i=1}^{t} \alpha_i} x_0 + \sqrt{1 - \prod_{i=1}^{t} \alpha_i} \epsilon_0 \tag{12}$$

$$= \sqrt{\bar{\alpha}_t} x_0 + \sqrt{1 - \bar{\alpha}_t} \epsilon_0 \tag{13}$$

$$\sim \mathcal{N}(x_t; \sqrt{\bar{\alpha}_t} x_0, (1 - \bar{\alpha}_t)I) \tag{14}$$

$$\tag{15}$$

Now, via Bayes' rule,

$$q(x_{t-1}|x_t, x_0) = \frac{q(x_t|x_{t-1}, x_0) q(x_{t-1}|x_0)}{q(x_t|x_0)} \tag{16}$$

$$= \frac{\mathcal{N}(x_t; \sqrt{\alpha_t} x_{t-1}, (1-\alpha_t)I) \, \mathcal{N}(x_{t-1}; \sqrt{\bar{\alpha}_{t-1}} x_0, (1-\bar{\alpha}_{t-1})I)}{\mathcal{N}(x_t; \sqrt{\bar{\alpha}_t} x_0, (1-\bar{\alpha}_t)I)} \tag{17}$$

$$\propto \exp\left\{ -\left[ \frac{(x_t - \sqrt{\alpha_t} x_{t-1})^2}{2(1-\alpha_t)} + \frac{(x_{t-1} - \sqrt{\bar{\alpha}_{t-1}} x_0)^2}{2(1-\bar{\alpha}_{t-1})} - \frac{(x_t - \sqrt{\bar{\alpha}_t} x_0)^2}{2(1-\bar{\alpha}_t)} \right] \right\} \tag{18}$$

$$= \exp\left\{ -\frac{1}{2}\left[ \frac{(x_t - \sqrt{\alpha_t} x_{t-1})^2}{1-\alpha_t} + \frac{(x_{t-1} - \sqrt{\bar{\alpha}_{t-1}} x_0)^2}{1-\bar{\alpha}_{t-1}} - \frac{(x_t - \sqrt{\bar{\alpha}_t} x_0)^2}{1-\bar{\alpha}_t} \right] \right\} \tag{19}$$

$$= \exp\left\{ -\frac{1}{2}\left[ \frac{-2\sqrt{\alpha_t} x_t x_{t-1} + \alpha_t x_{t-1}^2}{1-\alpha_t} + \frac{x_{t-1}^2 - 2\sqrt{\bar{\alpha}_{t-1}} x_{t-1} x_0}{1-\bar{\alpha}_{t-1}} + C(x_t, x_0) \right] \right\} \tag{20}$$

$$\propto \exp\left\{ -\frac{1}{2}\left[ \frac{-2\sqrt{\alpha_t} x_t x_{t-1}}{1-\alpha_t} + \frac{\alpha_t x_{t-1}^2}{1-\alpha_t} + \frac{x_{t-1}^2}{1-\bar{\alpha}_{t-1}} - \frac{2\sqrt{\bar{\alpha}_{t-1}} x_{t-1} x_0}{1-\bar{\alpha}_{t-1}} \right] \right\} \tag{21}$$

$$= \exp\left\{ -\frac{1}{2}\left[ \left( \frac{\alpha_t}{1-\alpha_t} + \frac{1}{1-\bar{\alpha}_{t-1}} \right) x_{t-1}^2 - 2\left( \frac{\sqrt{\alpha_t} x_t}{1-\alpha_t} + \frac{\sqrt{\bar{\alpha}_{t-1}} x_0}{1-\bar{\alpha}_{t-1}} \right) x_{t-1} \right] \right\} \tag{22}$$

Upon further expansion of Eq. 22,

$$= \exp\left\{-\frac{1}{2}\left[\frac{\alpha_t(1-\bar{\alpha}_{t-1})+1-\alpha_t}{(1-\alpha_t)(1-\bar{\alpha}_{t-1})}x_{t-1}^2 - 2\left(\frac{\sqrt{\alpha_t}x_t}{1-\alpha_t}+\frac{\sqrt{\bar{\alpha}_{t-1}}x_0}{1-\bar{\alpha}_{t-1}}\right)x_{t-1}\right]\right\} \tag{23}$$

$$= \exp\left\{-\frac{1}{2}\left[\frac{\alpha_t-\bar{\alpha}_t+1-\alpha_t}{(1-\alpha_t)(1-\bar{\alpha}_{t-1})}x_{t-1}^2 - 2\left(\frac{\sqrt{\alpha_t}x_t}{1-\alpha_t}+\frac{\sqrt{\bar{\alpha}_{t-1}}x_0}{1-\bar{\alpha}_{t-1}}\right)x_{t-1}\right]\right\} \tag{24}$$

$$= \exp\left\{-\frac{1}{2}\left[\frac{1-\bar{\alpha}_t}{(1-\alpha_t)(1-\bar{\alpha}_{t-1})}x_{t-1}^2 - 2\left(\frac{\sqrt{\alpha_t}x_t}{1-\alpha_t}+\frac{\sqrt{\bar{\alpha}_{t-1}}x_0}{1-\bar{\alpha}_{t-1}}\right)x_{t-1}\right]\right\} \tag{25}$$

$$= \exp\left\{-\frac{1}{2}\left(\frac{1-\bar{\alpha}_t}{(1-\alpha_t)(1-\bar{\alpha}_{t-1})}\right)\left[x_{t-1}^2 - 2\left(\frac{\sqrt{\alpha_t}x_t}{1-\alpha_t}+\frac{\sqrt{\bar{\alpha}_{t-1}}x_0}{1-\bar{\alpha}_{t-1}}\right)\frac{1-\bar{\alpha}_t}{(1-\alpha_t)(1-\bar{\alpha}_{t-1})}x_{t-1}\right]\right\} \tag{26}$$

$$= \exp\left\{-\frac{1}{2}\left(\frac{1-\bar{\alpha}_t}{(1-\alpha_t)(1-\bar{\alpha}_{t-1})}\right)\left[x_{t-1}^2 - 2\left(\frac{\sqrt{\alpha_t}x_t}{1-\alpha_t}+\frac{\sqrt{\bar{\alpha}_{t-1}}x_0}{1-\bar{\alpha}_{t-1}}\right)\frac{(1-\alpha_t)(1-\bar{\alpha}_{t-1})}{1-\bar{\alpha}_t}x_{t-1}\right]\right\} \tag{27}$$

$$= \exp\left\{-\frac{1}{2}\left(\frac{1}{(1-\alpha_t)(1-\bar{\alpha}_{t-1})}\cdot\frac{1-\bar{\alpha}_t}{1-\bar{\alpha}_t}\right)\left[x_{t-1}^2 - 2\frac{\sqrt{\alpha_t}(1-\bar{\alpha}_{t-1})x_t+\sqrt{\bar{\alpha}_{t-1}}(1-\alpha_t)x_0}{1-\bar{\alpha}_t}x_{t-1}\right]\right\} \tag{28}$$

$$\propto \mathcal{N}\left(x_{t-1}; \frac{\sqrt{\alpha_t}(1-\bar{\alpha}_{t-1})x_t+\sqrt{\bar{\alpha}_{t-1}}(1-\alpha_t)x_0}{1-\bar{\alpha}_t}, \frac{(1-\alpha_t)(1-\bar{\alpha}_{t-1})}{1-\bar{\alpha}_t}I\right) \tag{29}$$

From Eq. 29 it can be seen that,

$$\sigma_q^2(t) = \frac{(1-\alpha_t)(1-\bar{\alpha}_{t-1})}{(1-\bar{\alpha}_t)} \qquad \mu_q(t) = \frac{\sqrt{\alpha_t}(1-\bar{\alpha}_{t-1})x_t+\sqrt{\bar{\alpha}_{t-1}}(1-\alpha_t)x_0}{1-\bar{\alpha}_t}$$

Now further substituting $x_0 = \frac{x_t-\sqrt{1-\bar{\alpha}_t}\,\epsilon_0}{\sqrt{\bar{\alpha}_t}}$ in the mean term $\mu_q(t)$, we get,

$$\mu_q(t) = \frac{\sqrt{\alpha_t}(1-\bar{\alpha}_{t-1})x_t+\sqrt{\bar{\alpha}_{t-1}}(1-\alpha_t)x_0}{1-\bar{\alpha}_t} \tag{30}$$

$$= \frac{\sqrt{\alpha_t}(1-\bar{\alpha}_{t-1})x_t+\sqrt{\bar{\alpha}_{t-1}}(1-\alpha_t)\left(x_t-\frac{\sqrt{1-\bar{\alpha}_t}\,\epsilon_0}{\sqrt{\bar{\alpha}_t}}\right)}{1-\bar{\alpha}_t} \tag{31}$$

$$= \frac{\sqrt{\alpha_t}(1-\bar{\alpha}_{t-1})x_t+(1-\alpha_t)\left(x_t-\frac{\sqrt{1-\bar{\alpha}_t}\,\epsilon_0}{\sqrt{\alpha_t}}\right)}{1-\bar{\alpha}_t} \tag{32}$$

$$= \frac{\sqrt{\alpha_t}(1-\bar{\alpha}_{t-1})x_t}{1-\bar{\alpha}_t}+\frac{(1-\alpha_t)x_t}{(1-\bar{\alpha}_t)\sqrt{\alpha_t}}-\frac{(1-\alpha_t)\sqrt{1-\bar{\alpha}_t}\,\epsilon_0}{(1-\bar{\alpha}_t)\sqrt{\alpha_t}} \tag{33}$$

$$= \left(\frac{\sqrt{\alpha_t}(1-\bar{\alpha}_{t-1})}{1-\bar{\alpha}_t}+\frac{1-\alpha_t}{(1-\bar{\alpha}_t)\sqrt{\alpha_t}}\right)x_t-\frac{(1-\alpha_t)\sqrt{1-\bar{\alpha}_t}}{(1-\bar{\alpha}_t)\sqrt{\alpha_t}}\epsilon_0 \tag{34}$$

$$= \left(\frac{\alpha_t(1-\bar{\alpha}_{t-1})}{(1-\bar{\alpha}_t)\sqrt{\alpha_t}}+\frac{1-\alpha_t}{(1-\bar{\alpha}_t)\sqrt{\alpha_t}}\right)x_t-\frac{(1-\alpha_t)\sqrt{1-\bar{\alpha}_t}}{\sqrt{\alpha_t}}\epsilon_0 \tag{35}$$

$$= \frac{\alpha_t-\bar{\alpha}_t+1-\alpha_t}{(1-\bar{\alpha}_t)\sqrt{\alpha_t}}x_t-\frac{(1-\alpha_t)\sqrt{1-\bar{\alpha}_t}}{\sqrt{\alpha_t}}\epsilon_0 \tag{36}$$

$$= \frac{1-\bar{\alpha}_t}{(1-\bar{\alpha}_t)\sqrt{\alpha_t}}x_t-\frac{(1-\alpha_t)\sqrt{1-\bar{\alpha}_t}}{\sqrt{\alpha_t}}\epsilon_0 \tag{37}$$

$$= \frac{1}{\sqrt{\alpha_t}}x_t-\frac{(1-\alpha_t)\sqrt{1-\bar{\alpha}_t}}{\sqrt{\alpha_t}}\epsilon_0 \tag{38}$$

Similarly, $\mu_\theta(t) = \frac{1}{\sqrt{\alpha_t}}x_t-\frac{(1-\alpha_t)\sqrt{1-\bar{\alpha}_t}}{\sqrt{\alpha_t}}\epsilon_\theta(x_t,t)$

### 6.1.2 PROOF OF LEMMA-2

Let $x_0$ denote the true data. Thus, to increase the log-likelihood of the data we maximize the evidence lower bound as follows:

$$\ln p(x_0) = \ln \int p(x_{0:T}) \, dx_{1:T} \tag{39}$$

$$= \ln \int \frac{p(x_{0:T})}{q(x_{1:T}|x_0)} q(x_{1:T}|x_0) \, dx_{1:T} \tag{40}$$

$$= \ln \mathop{\mathbb{E}}_{q(x_{1:T}|x_0)} \left[ \frac{p(x_{0:T})}{q(x_{1:T}|x_0)} \right] \tag{41}$$

$$\overset{(a)}{\geq} \mathop{\mathbb{E}}_{q(x_{1:T}|x_0)} \left[ \ln \frac{p(x_{0:T})}{q(x_{1:T}|x_0)} \right] \tag{42}$$

$$= \mathop{\mathbb{E}}_{q(x_{1:T}|x_0)} \left[ \ln \frac{p(x_T) \prod_{t=1}^{T} p_\theta(x_{t-1}|x_t)}{\prod_{t=1}^{T} q(x_t|x_{t-1})} \right] \tag{43}$$

$$= \mathop{\mathbb{E}}_{q(x_{1:T}|x_0)} \left[ \ln \frac{p(x_T) p_\theta(x_0|x_1) \prod_{t=2}^{T} p_\theta(x_{t-1}|x_t)}{q(x_1|x_0) \prod_{t=2}^{T} q(x_t|x_{t-1})} \right] \tag{44}$$

$$\overset{(b)}{=} \mathop{\mathbb{E}}_{q(x_{1:T}|x_0)} \left[ \ln \frac{p(x_T) p_\theta(x_0|x_1) \prod_{t=2}^{T} p_\theta(x_{t-1}|x_t)}{q(x_1|x_0) \prod_{t=2}^{T} q(x_t|x_{t-1}, x_0)} \right] \tag{45}$$

$$= \mathop{\mathbb{E}}_{q(x_{1:T}|x_0)} \left[ \ln \frac{p_\theta(x_T) p_\theta(x_0|x_1)}{q(x_1|x_0)} + \ln \prod_{t=2}^{T} \frac{p_\theta(x_{t-1}|x_t)}{q(x_t|x_{t-1}, x_0)} \right] \tag{46}$$

$$\overset{(c)}{=} \mathop{\mathbb{E}}_{q(x_{1:T}|x_0)} \left[ \ln \frac{p(x_T) p_\theta(x_0|x_1)}{q(x_1|x_0)} + \ln \prod_{t=2}^{T} \frac{p_\theta(x_{t-1}|x_t)}{\frac{q(x_{t-1}|x_t, x_0) q(x_t|x_0)}{q(x_{t-1}|x_0)}} \right] \tag{47}$$

$$= \mathop{\mathbb{E}}_{q(x_{1:T}|x_0)} \left[ \ln \frac{p(x_T) p_\theta(x_0|x_1)}{q(x_1|x_0)} + \ln \frac{q(x_1|x_0)}{q(x_T|x_0)} + \ln \prod_{t=2}^{T} \frac{p_\theta(x_{t-1}|x_t)}{q(x_{t-1}|x_t, x_0)} \right] \tag{48}$$

$$= \mathop{\mathbb{E}}_{q(x_{1:T}|x_0)} \left[ \ln \frac{p(x_T) p_\theta(x_0|x_1)}{q(x_T|x_0)} + \sum_{t=2}^{T} \ln \frac{p_\theta(x_{t-1}|x_t)}{q(x_t - 1|x_t, x_0)} \right] \tag{49}$$

$$= \mathop{\mathbb{E}}_{q(x_{1:T}|x_0)} [\ln p_\theta(x_0|x_1)] + \mathop{\mathbb{E}}_{q(x_{1:T}|x_0)} \left[ \ln \frac{p(x_T)}{q(x_T|x_0)} \right] + \sum_{t=2}^{T} \mathop{\mathbb{E}}_{q(x_{1:T}|x_0)} \left[ \ln \frac{p_\theta(x_{t-1}|x_t)}{q(x_{t-1}|x_t, x_0)} \right] \tag{50}$$

$$\overset{(d)}{=} \mathop{\mathbb{E}}_{q(x_1|x_0)} [\ln p_\theta(x_0|x_1)] + \mathop{\mathbb{E}}_{q(x_T|x_0)} \left[ \ln \frac{p(x_T)}{q(x_T|x_0)} \right] + \sum_{t=2}^{T} \mathop{\mathbb{E}}_{q(x_t, x_{t-1}|x_0)} \left[ \ln \frac{p_\theta(x_{t-1}|x_t)}{q(x_{t-1}|x_t, x_0)} \right] \tag{51}$$

$$\overset{(e)}{=} \mathop{\mathbb{E}}_{q(x_1|x_0)} [\ln p_\theta(x_0|x_1)] - D_{KL}(q(x_T|x_0)||p(x_T)) \tag{52}$$

$$- \sum_{t=2}^{T} \mathop{\mathbb{E}}_{q(x_t|x_0)} [D_{KL}(q(x_{t-1}|x_t, x_0)||p_\theta(x_{t-1}|x_t))] \tag{53}$$

$$\overset{(f)}{\approx} - \sum_{t=2}^{T} \mathop{\mathbb{E}}_{q(x_t|x_0)} [D_{KL}(q(x_{t-1}|x_t, x_0)||p_\theta(x_{t-1}|x_t))] \tag{54}$$

Here $(a)$ holds via Jensen's inequality as $\log$ is a concave function. $(b)$ is true because additional conditioning doesn't affect the first-order Markovian transitional kernel $q(.|.)$. $(c)$ and $(e)$ are achieved via Bayes' rule while $(d)$ is true via marginalization property. As the first two terms in Eq. 52 are insignificant compared to the last term, so $(f)$ holds.

### 6.1.3 PROOF OF THEOREM-1

The variational divergence term in Eq. 2 is:

$$D_{KL}\left(Q(\theta)\middle\| Z \cdot \frac{P(\theta|D_f, D_r)}{P(D_f|\theta)}\right) = \mathop{\mathbb{E}}_{Q(\theta)}\left[\ln \frac{Q(\theta)P(D_f|\theta)}{Z \cdot P(\theta|D_f, D_r)}\right] \tag{55}$$

$$\stackrel{(g)}{=} \mathop{\mathbb{E}}_{Q(\theta)}\left[\ln \frac{Q(\theta)}{P(\theta|D_f, D_r)}\right] + \mathop{\mathbb{E}}_{Q(\theta)}\left[\ln P(D_f|\theta)\right] \tag{56}$$

$$\stackrel{(h)}{=} \mathop{\mathbb{E}}_{Q(\theta)}\left[\ln \frac{Q(\theta)}{P(\theta|D_f, D_r)}\right] + \mathop{\mathbb{E}}_{Q(\theta)}\left[\sum_{x_0 \in D_f} \ln P(x_0|\theta)\right] \tag{57}$$

$$= \underbrace{D_{KL}(Q(\theta)\|P(\theta|D_f, D_r))}_{A} + \underbrace{\mathop{\mathbb{E}}_{\theta \sim Q(\theta)}\left[\sum_{x_0 \in D_f} \ln P(x_0|\theta)\right]}_{B} \tag{58}$$

$(g)$ holds as the normalization constant is independent of $\theta$. $(h)$ is true because of the i.i.d. assumption on the data. Now, using the lemma presented below, we further expand the terms A and B in Eq. 58.

**Lemma 3** *The Kullback-Leibler divergence for two multivariate normal distributions is given by:*

$$D_{KL}(N(x; \mu_x, \Sigma_x)\|N(y; \mu_y, \Sigma_y)) = \frac{1}{2}\left[\log \frac{|\Sigma_y|}{|\Sigma_x|} - d + tr(\Sigma_y^{-1}\Sigma_x) + (\mu_y - \mu_x)^T \Sigma_y^{-1}(\mu_y - \mu_x)\right]$$

**Proof 2** *This proof can be found in any standard information theory textbook thus avoided here.*

Using the above Lemma 3, the KL-divergence in Lemma 2 can be written using Lemma 1 for $q(x_{t-1}|x_t, x_0) = \mathcal{N}(x_{t-1}; \mu_q(t), \sigma_q^2(t)I)$ with $\mu_q(t) = \frac{1}{\sqrt{\alpha_t}}x_t - \frac{1-\alpha_t}{\sqrt{1-\bar{\alpha}_t}\sqrt{\alpha_t}}\epsilon_0$ and $p_\theta(x_{t-1}|x_t) = \mathcal{N}(x_{t-1}; \mu_\theta(t), \sigma_q^2(t)I)$ with $\mu_\theta(t) = \frac{1}{\sqrt{\alpha_t}}x_t - \frac{1-\alpha_t}{\sqrt{1-\bar{\alpha}_t}\sqrt{\alpha_t}}\epsilon_\theta(x_t, t)$ where $\sigma_q^2(t) = \frac{(1-\alpha_t)(1-\bar{\alpha}_{t-1})}{(1-\bar{\alpha}_t)}$ as follows:

$$D_{KL}\big(q(x_{t-1}|x_t, x_0) \,\big\| \, p_\theta(x_{t-1}|x_t)\big) = D_{KL}\left(N(x_{t-1}; \mu_q, \Sigma_q(t)) \,\big\| \, N(x_{t-1}; \mu_\theta, \Sigma_q(t))\right) \tag{59}$$

$$= \frac{1}{2\sigma_q^2(t)}\left\|\frac{1}{\sqrt{\alpha_t}}x_t - \frac{1-\alpha_t}{\sqrt{1-\bar{\alpha}_t}\sqrt{\alpha_t}}\epsilon_\theta(x_t, t) - \frac{1}{\sqrt{\alpha_t}}x_t + \frac{1-\alpha_t}{\sqrt{1-\bar{\alpha}_t}\sqrt{\alpha_t}}\epsilon_0\right\|_2^2 \tag{60}$$

$$= \frac{1}{2\sigma_q^2(t)}\left\|\frac{1-\alpha_t}{\sqrt{1-\bar{\alpha}_t}\sqrt{\alpha_t}}\epsilon_0 - \frac{1-\alpha_t}{\sqrt{1-\bar{\alpha}_t}\sqrt{\alpha_t}}\epsilon_\theta(x_t, t)\right\|_2^2 \tag{61}$$

$$= \frac{1}{2\sigma_q^2(t)}\left\|\frac{1-\alpha_t}{\sqrt{1-\bar{\alpha}_t}\sqrt{\alpha_t}}(\epsilon_0 - \epsilon_\theta(x_t, t))\right\|_2^2 \tag{62}$$

$$= \frac{(1-\alpha_t)^2}{2\sigma_q^2(t)(1-\bar{\alpha}_t)\alpha_t}\|\epsilon_0 - \epsilon_\theta(x_t, t)\|_2^2 \tag{63}$$

$$= \frac{(1-\alpha_t)}{\alpha_t(1-\bar{\alpha}_{t-1})}\|\epsilon_0 - \epsilon_\theta(x_t, t)\|_2^2 \tag{64}$$

Now, the term A in Eq. 58 using Lemma 3 with the variational posterior distribution $Q(\theta) = \prod_{i=1}^d \mathcal{N}(\theta_i, \sigma_i^2)$ and the posterior distribution with full data $P(\theta|D_r, D_f) = \prod_{i=1}^d \mathcal{N}(\mu_i^*, \sigma_i^{*2})$ becomes:

$$D_{KL}(Q(\theta)\|P(\theta|D_f, D_r)) = \sum_{i=1}^d \left[\ln \frac{\sigma_i^*}{\sigma_i} + \frac{\sigma_i^2 + (\theta_i - \mu_i^*)^2}{2\sigma_i^{*2}} - \frac{1}{2}\right] \tag{65}$$

Building on the theoretical derivations above, the second term B in Eq. 58 can now be expressed using Monte Carlo estimation as follows:

$$\mathop{\mathbb{E}}_{\theta \sim Q(\theta)} \left[ \sum_{x_0 \in D_f} \ln P(x_0|\theta) \right] \approx \frac{1}{N} \sum_{m=1}^{N} \left[ \sum_{x_0 \in D_f} \ln P(x_0|\theta^m) \right] \tag{66}$$

$$\overset{(i)}{\gtrsim} \frac{1}{N} \sum_{m=1}^{N} \left[ - \sum_{x_0 \in D_f} \sum_{t=2}^{T} \mathop{\mathbb{E}}_{q(x_t|x_0)} \left[ D_{KL}(q(x_{t-1}|x_t, x_0)||p_\theta(x_{t-1}|x_t)) \right] \right] \tag{67}$$

$$\overset{(j)}{\approx} - \sum_{x_t \in D_f} \sum_{t=2}^{T} \frac{(1 - \alpha_t)}{\alpha_t(1 - \bar{\alpha}_{t-1})} ||\epsilon_0 - \epsilon_\theta(x_t, t)||^2 \tag{68}$$

$(i)$ holds as a consequence of Lemma 2. Now using a crude estimate of $N = 1$ the $(j)$ holds by equation 64. Finally, incorporating both A and B terms using Eq. 65 and Eq. 68 respectively we get,

$$D_{KL}\left( Q(\theta) \middle|\middle| Z \cdot \frac{P(\theta|D_f, D_r)}{P(D_f|\theta)} \right) \gtrsim \underbrace{- \sum_{x_t \in D_f} \sum_{t=2}^{T} \frac{(1 - \alpha_t)}{\alpha_t(1 - \bar{\alpha}_{t-1})} ||\epsilon_0 - \epsilon_\theta(x_t, t)||^2}_{\text{A}}$$

$$+ \underbrace{\sum_{i=1}^{d} \left[ \frac{(\theta_i - \mu_i^*)^2}{2\sigma_i^{*2}} + \frac{\sigma_i^2}{2\sigma_i^{*2}} + \log \frac{\sigma_i^*}{\sigma_i} - \frac{1}{2} \right]}_{\text{B}}$$

## 6.2 IMPLEMENTATION DETAILS

### 6.2.1 DATASETS AND MODELS

- **MNIST:** The MNIST dataset consist of $28 \times 28$ grayscale representing handwritten digits from 0 to 9. The MNIST dataset contains 60,000 training images and 10,000 testing images. We have used the same architectural model detailed in the open-source implementation: https://github.com/explainingai-code/DDPM-Pytorch (Ronneberger et al., 2015) for the MNIST dataset.

- **CIFAR-10:** CIFAR10 consists of 60,000 $32 \times 32$ color images distributed across 10 classes with 6000 images in each class. We adopt an unconditional DDPM model based on the approach by (Nichol & Dhariwal, 2021), utilizing their official implementation provided in https://github.com/openai/improved-diffusion

### 6.2.2 INITIAL TRAINING

- **MNIST:** For the pre-training of the DDPM model on MNIST, we adopt the hyperparameters from the open source code base mentioned above. Specifically, a randomly initialized DDPM model is pre-trained on the train set of MNIST and optimized using a learning rate of $10^{-4}$ for 40 epochs with a batch size of 64. The diffusion process follows a noise scheduling strategy where the noise variance $\alpha_1$ at $t = 1$ is set to 0.0001, and it linearly increases to $\alpha_T = 0.02$ at the final time-step $T$=1000. We have trained 5 models to get the model mean $\mu_i^*$ and variance $\sigma_i^*$ parameter.

- **CIFAR-10:** We use a unconditional DDPM checkpoint, pre-trained on CIFAR-10, from https://github.com/openai/improved-diffusion. We fine-tune this model on the train set of CIFAR-10 for 90k iterations, with a batch size of 128, and a learning rate of $10^{-5}$. The total diffusion steps $T$ is set to 4000 with a cosine noise scheduling strategy. Here We have trained 4 models to get the model mean $\mu_i^*$ and variance $\sigma_i^*$ parameters. For the pre-training of both DDPM models, we use Adam optimizer.

### 6.2.3 UNLEARNING

After pre-training the DDPM models on their respective datasets, as described earlier, we now outline the unlearning process for each. We optimize using our proposed loss function, as defined in Eq. 3, ensuring effective feature removal while maintaining model performance as,

1. **MNIST:** The model is optimized for unlearning over only 1 epoch using the Adam optimizer with a learning rate of $10^{-6}$ and a batch size of 128. We set the total timesteps $T$ to 1000, with the noise scheduler parameters $\alpha_1 = 0.0001$ and $\alpha_T = 0.02$.
2. **CIFAR10:** Similar optimization parameters are used as for MNIST, but the model is trained over only 4 epochs. Here, we adjust the total timesteps to $T = 500$, with $\alpha_1 = 0.0002$ and $\alpha_T = 0.04$.

For all of the benchmark datasets, we select and report the results for $\gamma \in \{0.1, 0.3, 0.6, 0.8\}$.

## 6.3 ADDITIONAL VISUAL RESULTS

### 6.3.1 UNLEARNED SAMPLES USING VDU

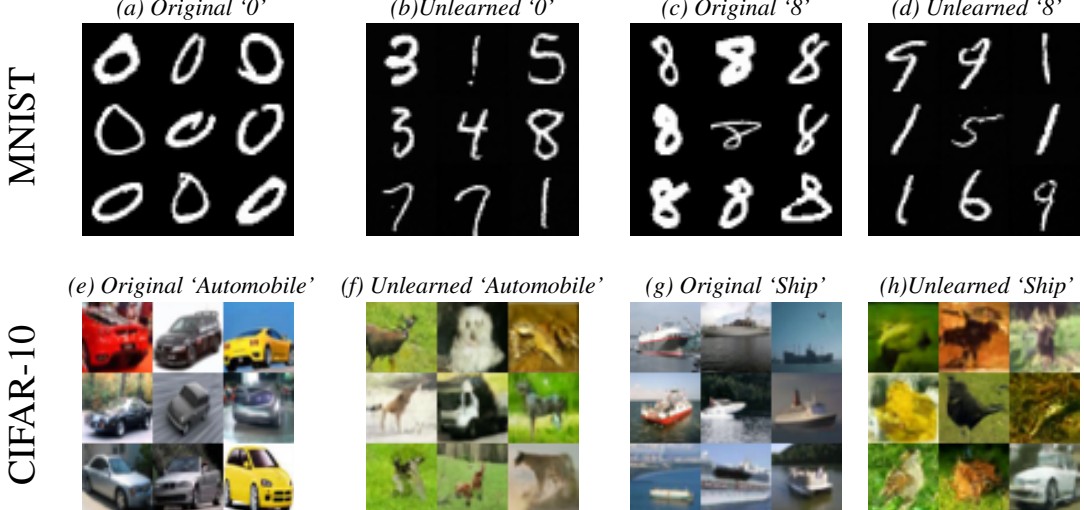

Figure 4: Generated samples from the pre-trained DDPM model and different class unlearned models. (a), (c), (e), and (g) are generated using the pre-trained model while (b), (d), (f), and (h), are generated from the corresponding unlearned models. Our method shows superior image quality after unlearning.

### 6.3.2  GENERATED SAMPLES USING SELECTIVE AMNESIA (SA)

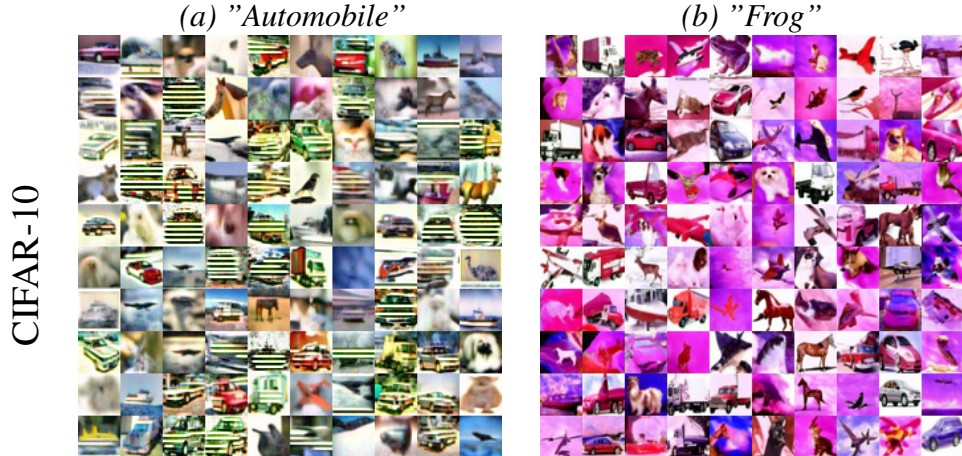

Figure 5: Generated samples from the unlearned model using SA method on CIFAR-10. This method shows poor image quality after unlearning for a few epochs.

