# OpenReview forum: "Variational Diffusion Unlearning: a variational inference framework for unlearning in diffusion models"
_NeurIPS.cc/2024/Workshop/SafeGenAi — SafeGenAi Poster_

### Official Review · Reviewer_7QH6 · 2024-10-09
**The paper discusses an approach to unlearn subset of training data on a pretrained diffusion model.**

**Rating:** 7
**Confidence:** 4

**Review:**

This work addresses a significant challenge of unlearning in generative models, particularly within the context of diffusion models. Unlearning is essential in large-scale models where retraining from scratch, by excluding a subset of data for various reasons, which can be highly inefficient.

The authors propose a variational inference-based method to facilitate data unlearning. They provide a comprehensive derivation of their approach, detailing how it operates. While the methodology is influenced by prior work (e.g., Nguyen et al., 2020b), its adaptation and necessary modifications for application in diffusion models are notable.

The authors conducted extensive experiments to validate the effectiveness of their proposed approach in unlearning, showcasing its potential in practical scenarios.

---

### Official Review · Reviewer_VVB9 · 2024-10-09
**This paper presents a method that leverages Variational inference to conduct unlearning in diffusion models. They provide promising preliminary results demonstrating that there approach can successfully prevent a model from producing part of a training distribution whilst retaining the rest.**

**Rating:** 7
**Confidence:** 5

**Review:**

This paper is of high quality and very clear making it easy to follow and replicate. It is particularly original as I have not seen techniques from probalistic modelling used to conduct safety research in this way.

Pros:
- This is an excellent use of varitional inference, the authors set the problem up in a very elegant way, and provide thorough theoretical justification for their choices.
- The method clearly works very well in both MNIST and CIFAR.
- Provides a promising and novel approach to unlearning which is distinguished from current work by it's deep grounding in variational inference.

Cons:
- The method could be improved by allowing the user to unlearn a specific feature in favour of another. For instance in MNIST, if the model could be instructed to produce 5s instead of 1s.
- The authors could have benchmarked heir approach against more already existing methods from the literature for example: Li, Guihong, Hsiang Hsu, and Radu Marculescu. "Machine unlearning for image-to-image generative models." arXiv preprint arXiv:2402.00351 (2024).
- A few points on linguistic and mathematical clarity: Line 191 into 192 needs to be reworded for grammar. In equation 1, if Dr and Df are conditionally independent given \theta. then P(Dr, Df|\theta)P(\theta) = P(Df|\theta)P(Dr|\theta)P(\theta) not proportional to (unless the proportionality factor is 1). Line 267 - modify Let's to Let.